# Assessment of energy expenditure during high intensity cycling and running using a heart rate and activity monitor in young active adults

**Malgorzata Klass**[1]*, **Vitalie Faoro**[2], **Alain Carpentier**[1]

**1** Laboratory for Biometry and Exercise Nutrition, Université Libre de Bruxelles (ULB), Brussels, Belgium,
**2** Cardiopulmonary Exercise Laboratory, Université Libre de Bruxelles (ULB), Brussels, Belgium

* mklass@ulb.ac.be

## Abstract

### Objective

Although high intensity physical activities may represent a great proportion of the total energy expenditure in active people, only sparse studies have investigated the accuracy of wearable monitors to assess activity related energy expenditure (AEE) during high intensity exercises. Therefore, the purpose of the present study was to investigate the accuracy of the Actiheart, a light portable monitor estimating AEE based on heart rate (HR) and activity counts (ACT), during two popular activities (running and cycling) performed at high intensities. The benefit of an individual calibration of the HR-AEE relationship established during a preliminary maximal test was also evaluated.

### Methods

AEE was estimated in eighteen active adults (4 women and 14 men; 25 ± 4 yr) with indirect calorimetry using a respiratory gas analysis system (reference method) and the Actiheart during 5-min running and cycling at 60, 75 and 85% of maximal oxygen uptake (VO$_2$max) previously determined during a maximal test performed on a treadmill or cycle ergometer. For the Actiheart, AEE was estimated either using the group or individual calibrated equations available in the dedicated software, and their respective HR, ACT or combined HR/ACT algorithms.

### Results

When the HR algorithm was used for cycling and the HR or HR/ACT algorithms for running, AEE measured by the Actiheart increased proportionally to exercise intensity from 60 to 85% VO$_2$max (P<0.001). Compared to indirect calorimetry, the Actiheart group calibrated equations slightly to moderately underestimated (3 to 20%) AEE for the three exercise intensities (P<0.001). Accuracy of AEE estimation was greatly improved by individual calibration of the HR-AEE relationship (underestimation below 5% and intraclass correlation coefficient [ICC]: 0.79–0.93) compared to group calibration (ICC: 0.64–0.79).

**Data Availability Statement:** All relevant data are within the manuscript and its Supporting Information files.

**Funding:** The authors received no specific funding for this work.

**Competing interests:** The authors have declared that no competing interests exist.

## Conclusion

The Actiheart enables to assess AEE during high intensity running and cycling when the appropriate algorithm is applied. Since an underestimation was present for group calibration, an individual and sport-specific calibration should be performed when a high accuracy is required.

## Introduction

Quantifying precisely the activity related energy expenditure (AEE) is necessary to check the adequacy with physical activity recommendations in the general population and is crucial for athletes in whom AEE represents up to 70% of total energy expenditure [1]. Currently, the gold standard to assess AEE is indirect calorimetry (IC) which quantifies AEE by measuring the oxygen consumed ($VO_2$) and the carbon dioxide released ($VCO_2$). The method is however impractical in the field, and even the portable IC devices are cumbersome and restrict the mobility [2]. Accurate portable methods, more applicable to field conditions, would thus enable determining more precisely athletes' training volume and ensure that the dietary intake covers the actual energy expenditure.

Several light wearable monitors have been developed which record and convert accelerations and/or heart rate (HR) to AEE based on proprietary algorithms. Monitors integrating accelerometers counts body movements in one or three axes and can provide valid estimation of AEE during activities mainly composed of level ambulation [3]. They have however a limited ability to assess AEE during most sport activities, particularly at higher intensities [2,4–7].

HR is known to increase linearly with $VO_2$ during moderate to high intensity activities [8] and may be used to predict associated energy expenditure [9]. AEE and the slope of the relationship between the increase in HR and AEE depends on exercise type and factors such as fitness level, work efficiency, age, sex, height, weight, body composition and hormonal status [10–12]. Group calibrated equations, available in most wearable monitors, integrates factors such as subject's age, sex, height and weight but not most of the other parameters influencing AEE [13,14]. Therefore, an individualized calibration is recommended to improve the accuracy of AEE prediction at the individual level [15], and may be performed in some research-grade devices (i.e. Actiheart).

Previous studies have suggested that the combination of HR and activity (movement counts) monitoring may overcome the limitations of each technique used separately [16–21]. Nowadays, some miniaturized consumer devices and research-grade devices (i.e. Actiheart) record simultaneous HR and activity to quantify AEE. Even though consumer devices are more affordable, and some of them provide reasonably good estimates of energy expenditure, their proprietary algorithms are generally unknown and their accuracy is considered as insufficient to be used for precise guidance on training volume and energy balance [14,22]. Amongst the research-grade devices, the Actiheart (AH) has been shown to be superior to consumer devices to quantify energy expenditure during daily activities, during walking and moderate intensity running and cycling [13,22]. The AH is a compact and light chest-worn device that combines recording of HR and accelerations at a high resolution. A dedicated software allows to extract and analyze the HR and accelerations data and uses them concomitantly or separately to calculate energy expenditure. The AH has previously been validated against doubly labeled water and IC during low to moderate intensity activities [20,22–25]. Surprisingly, while higher intensity activities may represent a great proportion of the total AEE in active

people, only sparse studies have investigated the validity of the AH during higher intensity running exercises [26,27]. In addition, although cycling is frequently practiced by recreational and high-level athletes, to our knowledge, no study has analyzed the accuracy of the AH to estimate AEE during higher intensity cycling exercises.

Therefore, the aim of the present study is to investigate the accuracy of the AH during cycling and running exercises ranging from 60 to 85% of VO₂max in recreational athletes and to evaluate the benefit of an individual calibration of the HR-AEE relationship using a maximal incremental test. We hypothesized that the combination of HR and activity counts recording would enable AEE estimation during both high intensity running and cycling exercises, and that individual calibration would improve accuracy at the individual level.

## Materials and methods

### Participants

Twenty healthy subjects took part in the study. Poor HR signal has generated unreadable HR data in two subjects. Finally, data of eighteen subjects (four women and fourteen men; age: 25 ± 4 yr; height: 1.77 ± 0.10 m; mass: 70.9 ± 11.7 kg) were analyzed. To take part in the study subjects had to be between 18 and 40 years old, be in good health, be free of any contraindication to the practice of sport and have a sufficient fitness level to run and cycle at high intensities. The experimental protocol was approved by the Hospital-Faculty Ethics Committee of Erasme-ULB (approval P2018/093). Prior to their participation, all volunteers received oral and written information regarding the nature and purpose of the study and signed an informed consent form. All the subjects were staff members or students at the faculty of Motor Sciences (Université libre de Bruxelles, ULB). To estimate their activity level, we ask them which sport(s) they practice and how many times a week. They reported to get involved three to six times a week in various sport activities (running, cycling, football, basketball, rugby, gymnastics and aerobics classes).

### Protocol

All the subjects took part in four experimental sessions spaced by 72 h to one week and conducted for each subject at about the same time in the morning or the afternoon. Two sessions were dedicated to the maximal incremental running and cycling tests (sessions 1 and 3; Fig 1) performed respectively on a treadmill (Pulsar 3p; h/p/Cosmos, Nussdorf, Germany) or a cycle ergometer (Ergoselect II 1200; Ergoline, Bitz, Germany). During the two other sessions, subjects ran or cycled during 5-min stages at three intensities (sessions 2 and 4; Fig 1). The ergometer used for the maximal test during the first session was drawn by lot. During the second session, subjects performed the 5-min stages on the same ergometer. Session three and four were dedicated respectively to the maximal test and 5-min stages on the second ergometer (Fig 1). Subjects were asked not to take part in any sport activity within the 24 h prior to the experimental sessions and to refrain from food and caffeine intake within the 3 h before the experiment.

### Sleeping HR

Since sleeping HR varies from subject to subject and is lower in fit subjects [9,10], the group calibrated HR algorithm of the AH (see below for further details) integrates the HR above sleeping HR (HRaS), instead of raw HR, to reduce the error due to intersubject variance in the HR–AEE relationship in the calculation of AEE [9]. Since we were not able to measure sleeping HR directly, it was estimated from the lying HR using the following equation: 0.83*lying

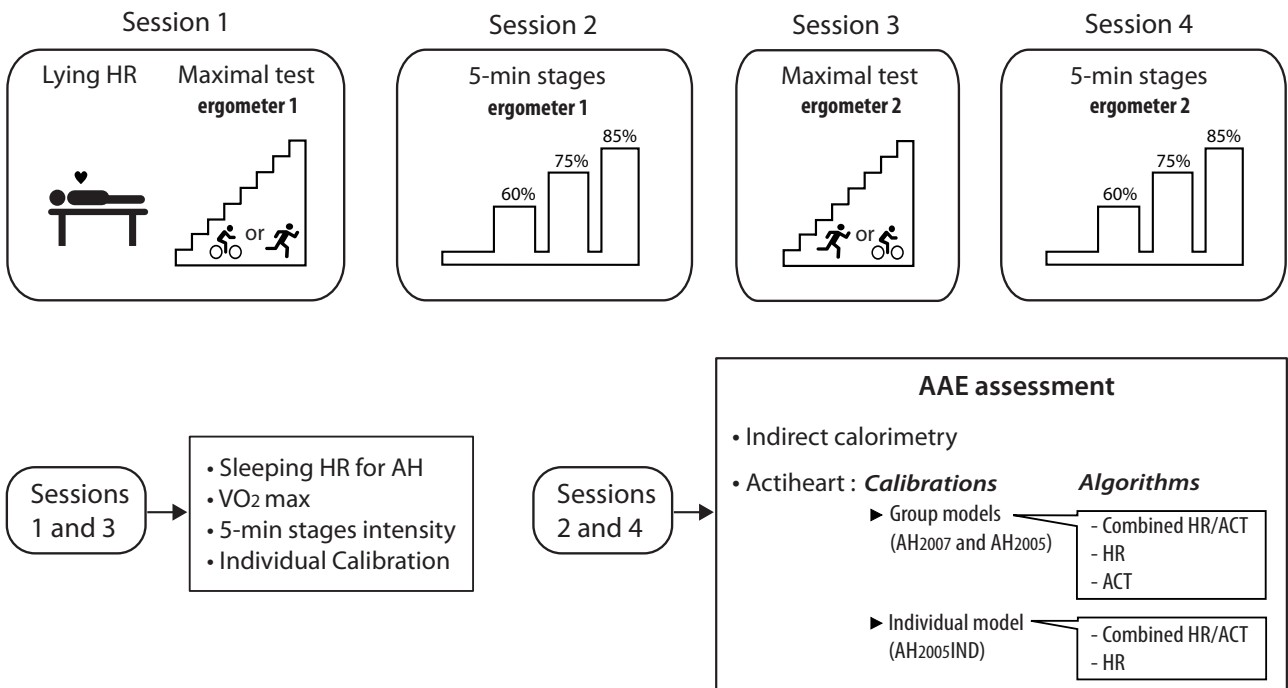

**Fig 1. Experimental protocol.** All subjects took part in four experimental sessions. Sessions 1 and 3 were dedicated to the recording of lying heart rate (HR) and maximal test on a treadmill or cycle ergometer in a randomized order. Lying HR was used to set sleeping HR in the AH software. Maximal tests allowed to determine: VO₂max, intensities for the 5-min stages and individual calibration of the HR-AEE relationship for the AH. During sessions 2 and 4, subjects performed 5-min stages at 60, 75, 85% VO₂max, while HR and activity counts (ACT) were recorded by the AH and gas exchange by indirect calorimetry. Before and between the stages, subjects walked at 3.5 km.h⁻¹ or cycled at 60 W. AEE was then calculated offline using the Weir equation for indirect calorimetry, and the group calibrated models (AH$_{2005}$ and AH$_{2007}$) or the individually calibrated model (AH$_{2005}$IND) and their respective algorithms (ACT, HR, combined HR/ACT) for the AH.

HR [18,23,24,28]. Lying HR was measured at the beginning of the first session (Fig 1). To that aim subjects had to lie down, eyes closed, during 10 min in a quiet place while their HR was measured using the AH and the average HR from minute 5 to 9 was then calculated [28].

## Maximal incremental tests

After a familiarization with the procedures, subjects performed an incremental maximal test on the treadmill or cycle ergometer (Fig 1). The maximal cycling test started at a workload of 30 W for women and 60 W for men, and was increased by 20 W (women) or 30 W (men) every minute until volitional exhaustion, i.e. the moment the subject could not maintain a pedaling cadence of 60 rpm. During the maximal running test, subjects started with a 3-min warm-up at a speed of 7 km.h⁻¹ for women and between 7 and 9 km.h⁻¹ for men depending on their preference. Thereafter, speed was increased every minute by 1 km.h⁻¹ until the subject was unable to maintain the imposed running speed. To compensate the lack of air resistance, the inclination of the treadmill was set at 1% [29]. During both tests, subjects were wearing a facial mask connected to the HypAir Professional respiratory gas analysis system (Medisoft, Dinant, Belgium) to measure breath by breath VO₂ and VCO₂. HR was measured continuously by a cardio frequency meter (Polar H1, Kempele, Finland) and transferred to the Medisoft software installed on the PC using a Heart Rate Monitor Interface (HRMI, SparkFun Electronics, Niwot, Colorado, USA). Before each test, the system was calibrated with room air and standardized gas mixture (16% O₂ and 4% CO₂) and calibrated for volume. For each

subject, data recorded during the maximal tests were used to determine VO$_2$max, maximal HR, workloads/speeds for each 5-min stage and individual calibration of the AH.

### Five-minute stages

During the sessions dedicated to the 5-min stages, subjects were requested to run or cycle at intensities corresponding to 60, 75 and 85% of their VO$_2$max determined during the maximal test on the same ergometer (Fig 1). They were wearing the facial mask connected to the respiratory gas analysis system and the AH during all exercises. Before the first stage, subjects walked at 3.5 km.h$^{-1}$ or cycled at 60 W during 5 min to verify the stability of the recordings, then the speed/load was increased to 60, 75 and 85% VO$_2$max. Between the stages, the intensity was lowered to 3.5 km.h$^{-1}$ or 60 W for one minute to allow a short recovery and a clear distinction between stages to facilitate further analysis.

### Energy expenditure assessment

AEE was quantified during running and cycling using indirect calorimetry (IC) and the 4[th] version of the AH device (CamNtech limited, Cambridge, UK). To allow a stabilization of gas exchange, the 2 last minutes of each 5-min running and cycling stage were analyzed [6].

IC was used as the reference method to quantify energy expenditure during the efforts. Firstly, VO$_2$ and CO$_2$ were measured using the respiratory gas analysis system described above. The total energy expenditure (TEE) was then calculated using the Weir equation [30]:

$$TEE\,(kcal.min^{-1}) = 3,9^* VO_2\,(L.min^{-1}) + 1,1^* VCO_2\,(L.min^{-1})$$

To determine AEE, resting energy expenditure (estimated by the AH software using the Schofield equation [31]) was calculated for each subject (based on sex, age and weight) and subtracted from the TEE.

For each subject, age, sex, weight, height and sleeping heart rate were entered in the AH software and the AH was set-up for short term recording (using a 15-s recording epoch). It was thereafter placed on subject chest, according to manufacturer's instructions, using two adhesive ECG electrodes. Shortly, one extremity of the device was placed just below the apex of the sternum and the other at the V4 or V5 position. It recorded HR data and uniaxial accelerations, converted into activity counts (ACT), during the 5-min stages.

### Actiheart data analysis

At the end of the four 5-min stages, HR data and activity counts (ACT) recorded by the accelerometer were downloaded from the device and analyzed offline using the manufacturer software (version 4.0.11) to estimate AEE. By default, the software proposes different versions of group calibrated models that use HR or ACT algorithms proposed by Brage et al. [20,32]. In addition to HRaS and ACT data, subject's sex, considered as an important predictor of AEE [10,11], is included as a variable in those algorithms. The user can choose to use either the HR or ACT algorithm separately or a branched equation [19] that enables to combine them (HR/ACT algorithm). When the branched equation is selected, the relative contribution of ACT and HR algorithms to the calculation of AEE is weighted according to specific ACT and HR thresholds: when both ACT and HR values are low, ACT algorithm has more weight, whereas when ACT and HR values are high, HR algorithm is the predominant contributor to AEE estimation [19,23]. If preferred to the default group calibrated models, an individual calibration of the HR-AEE relationship can be performed, using either the step test built in the software or another exercise test.

In the present study, we used the two main group calibrated models available for adults in the AH software: the original version presented by Brage et al. in 2005 [20] ($AH_{2005}$; termed *"Group ACT/Group HR (old)"* in the AH software) and the latest version adapted by Brage et al. in 2007 [32] ($AH_{2007}$; termed *"Group Cal JAP2007"* in the AH software). Each model uses its own HR and ACT algorithms and a common branched equation described in the AH user manual. The $AH_{2007}$ model is currently recommended by the manufacturer. We therefore wanted to test if it really improves AEE estimation during running and cycling compared to $AH_{2005}$. To assess the additional benefit of an individual calibration of the HR-AEE relationship, we used the relation established between AEE and HR data collected during the last 10 s of each stage of the maximal cycling and running tests. Since individual calibration is only possible with $AH_{2005}$, the individually calibrated model ($AH_{2005}IND$; termed *"Group ACT/Ind HR (old)"* in the AH software) shares the group-calibrated ACT algorithm of $AH_{2005}$ but the HR algorithm uses the individual HR-AEE relationship determined during the maximal incremental tests.

## Statistical analysis

The normality of the data was firstly controlled using a Shapiro-Wilk normality test. A two-factor (AEE estimation method x exercise intensity) ANOVA with repeated measures was used to analyze the AEE measured using IC and the different AH models during running and cycling exercises performed at 60, 75 and 85% $VO_2max$. When a significant main effect was found, a Bonferroni's post hoc test was used to compare selected data points. More specifically, for each exercise intensity, the AEE estimated by each of the AH models ($AH_{2007}$, $AH_{2005}$ and $AH_{2005}IND$), using the combined HR/ACT, HR or ACT algorithm, was compared to IC. The sensitivity of the AH to increases in exercise intensity was analyzed by comparing the AEE between successive stages.

For each exercise intensity, a Bland-Altman analysis was performed to calculate the mean bias and the 95% limits of agreement (LoA) between AEE measured by IC and the different AH models and algorithms. The strength of the agreement was assessed by intraclass correlation coefficient (ICC) estimates, and their 95% confidence intervals, calculated using a two-way mixed-effects model, single-measurement and absolute-agreement [33]. Based on the 95% confidence interval of the ICC, values less than 0.5, between 0.5 and 0.75, between 0.75 and 0.9, and greater than 0.90 indicate respectively poor, moderate, good, and excellent agreement [33]. Analyses were performed using GraphPad Prism software version 6 (La Jolla, California, USA) and MedCalc statistical software Version 18.11.3 (Ostend, Belgium). For all comparisons, the statistical level of significance was set at 0.05.

## Results

Participants performance characteristics during the maximal running and cycling tests, as well as during the last 2 min of the 5-min stages at 60, 75 and 85% $VO_2max$, are reported in Table 1.

The ANOVA for AEE during the running and cycling exercises of different intensities revealed a main effect for the method, exercise intensity, and an interaction between method and exercise intensity (P<0.001). The analysis of AEE estimated by the AH during stages of increasing intensity, revealed that group and individually calibrated models are sensitive to changes in exercise intensity (Fig 2). Indeed, during the running exercise, regardless of the algorithm, AEE estimated by the AH increased significantly, and proportionally to the increase in exercise intensity, from 60 to 75% $VO_2max$ and from 75 to 85% $VO_2max$ (post-tests P-values <0.01). This was also true for the cycling exercise when HR/ACT and HR algorithms were

**Table 1. Participants performance characteristics during the maximal tests and 5-min stages.**

|  | Running | Cycling |
|---|---|---|
| $VO_2max$ (ml.kg$^{-1}$.min$^{-1}$) | 51.4 [6.6] | 46.4 [7.9] |
| HR max (bpm) | 192 [8] | 185 [7] |
| Maximal speed (km.h$^{-1}$) or load (W) | 16.0 [1.7] | 266 [60] |
| Speed (km.h$^{-1}$) or load (W) at |  |  |
| 60% $VO_2max$ | 9.4 [1.5] | 142 [41] |
| 75% $VO_2max$ | 11.4 [1.5] | 177 [44] |
| 85% $VO_2max$ | 13.3 [1.5] | 209 [49] |

Mean values and [SD]. Abbreviations: HR, heart rate; $VO_2max$, maximal oxygen uptake.

used (P-values <0.001). However, for the ACT algorithm, AEE increased only very slightly from 60 to 85% $VO_2max$ (P-values <0.01).

For the running exercise, when HR/ACT or HR algorithm was used, post hoc tests showed a significant difference between AEE estimated by IC and by the AH group calibrated models for all exercise intensities (P-values <0.001; Fig 2A and 2B). Similarly, the ACT algorithm underestimated AEE (P-values <0.001), except for AH$_{2007}$ at 60% $VO_2max$ (P = 0.83, Fig 2C). Individual calibration (AH$_{2005}$IND) improved AEE estimation for both HR/ACT and HR algorithms. There was indeed no more significant difference with IC for 75 and 85% $VO_2max$ (P-values range: 0.06–0.99) and a very slight difference for 60% $VO_2max$ (P-values <0.01; Fig 2A and 2B, and bias in Table 2).

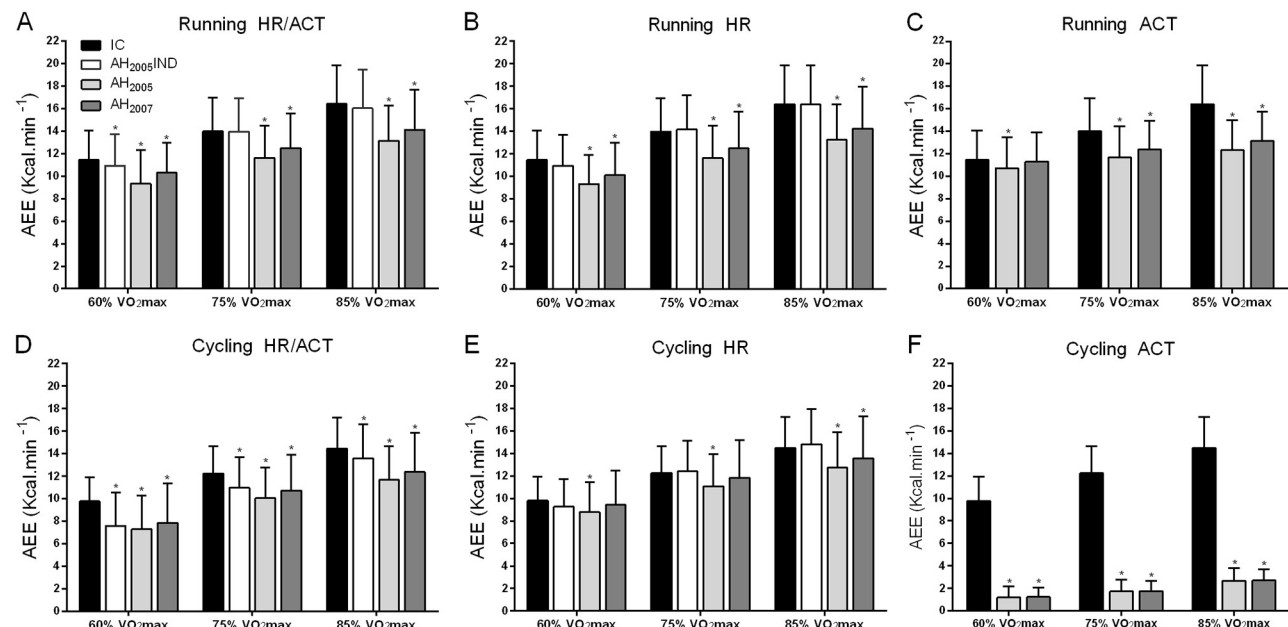

**Fig 2. Activity related energy expenditure (AEE) during running and cycling measured by indirect calorimetry (IC) and the different Actiheart models (AH$_{2007}$, AH$_{2005}$, AH$_{2005}$IND) and the activity counts (ACT), heart rate (HR) or combined HR/ACT algorithms.** For the HR/ACT and HR algorithms, AEE estimated by the Actiheart increased significantly from 60 to 85% $VO_2max$ during running and cycling (P <0.01). For the ACT algorithm, AEE also increased with intensity increments during running (P <0.01), but only very slightly and not proportionally to intensity during cycling. *P <0.05 between AEE measured by the Actiheart and indirect calorimetry.

**Table 2. Agreement between activity related energy expenditure measured by the Actiheart and indirect calorimetry quantified by ICC estimates (left panel) and bias (right panel).**

| | ICC and 95% confidence interval | | | | | | Mean bias [SD] and 95% LoA (kcal.min$^{-1}$) | | | | | |
| | $AH_{2007}$ | | $AH_{2005}$ | | $AH_{2005}IND$ | | $AH_{2007}$ | | $AH_{2005}$ | | $AH_{2005}IND$ | |
|---|---|---|---|---|---|---|---|---|---|---|---|---|
| **Running** | | | | | | | | | | | | |
| HR/ACT | | | | | | | | | | | | |
| 60% $VO_2$max | 0.78 | (0.32, 0.92) | 0.65 | (-0.07, 0.89) | 0.92 | (0.77, 0.97) | -1.2 [1.5] | (-4.0, 1.7) | -2.1 [1.6] | (-5.3, 1.0) | -0.5 [1.0] | (-2.5, 1.6) |
| 75% $VO_2$max | 0.79 | (0.17, 0.93) | 0.68 | (-0.08, 0.91) | 0.93 | (0.81, 0.97) | -1.5 [1.5] | (-4.4, 1.4) | -2.3 [1.3] | (-4.9, 0.2) | 0.0 [1.2] | (-2.3, 2.3) |
| 85% $VO_2$max | 0.76 | (-0.06, 0.94) | 0.63 | (-0.05, 0.90) | 0.93 | (0.83, 0.97) | -2.3 [1.4] | (-5.1, 0.5) | -3.3 [1.2] | (-5.6, -0.9) | -0.4 [1.2] | (-2.8, 2.0) |
| HR | | | | | | | | | | | | |
| 60% $VO_2$max | 0.74 | (0.24, 0.91) | 0.64 | (-0.08, 0.89) | 0.92 | (0.77, 0.97) | -1.4 [1.6] | (-4.6, 1.9) | -2.1 [1.4] | (-5.0, 0.7) | -0.5 [1.0] | (-2.4, 1.4) |
| 75% $VO_2$max | 0.79 | (0.21, 0.93) | 0.68 | (-0.08, 0.92) | 0.92 | (0.80, 0.97) | -1.5 [1.6] | (-4.5, 1.8) | -2.4 [1.3] | (-4.9, 0.1) | 0.2 [1.2] | (-2.2, 2.6) |
| 85% $VO_2$max | 0.77 | (-0.03, 0.94) | 0.64 | (-0.06, 0.91) | 0.93 | (0.82, 0.97) | -2.2 [1.5] | (-5.2, 0.8) | -3.2 [1.2] | (-5.5, -0.8) | 0.0 [1.3] | (-2.6, 2.6) |
| ACT | | | | | | | | | | | | |
| 60% $VO_2$max | 0.81 | (0.55, 0.92) | 0.81 | (0.55, 0.93) | --- | --- | -0.2 [1.7] | (-3.4, 3.1) | -0.7 [1.5] | (-3.7, 2.3) | --- | --- |
| 75% $VO_2$max | 0.72 | (0.10, 0.91) | 0.66 | (-0.09, 0.90) | --- | --- | -1.6 [1.6] | (-4.7, 1.5) | -2.3 [1.5] | (-5.2, 0.5) | --- | --- |
| 85% $VO_2$max | 0.53 | (-0.21, 0.92) | 0.45 | (-0.15, 0.90) | --- | --- | -3.3 [1.8] | (-6.8, 0.3) | -4.1 [1.7] | (-7.5, -0.7) | --- | --- |
| **Cycling** | | | | | | | | | | | | |
| HR/ACT | | | | | | | | | | | | |
| 60% $VO_2$max | 0.51 | (0.04, 0.79) | 0.47 | (-0.09, 0.79) | 0.51 | (-0.06, 0.81) | -1.9 [2.6] | (-6.9, 3.1) | -2.5 [2.1] | (-6.6, 1.7) | -2.2 [2.1] | (-6.2, 1.9) |
| 75% $VO_2$max | 0.64 | (0.17, 0.86) | 0.55 | (-0.07, 0.84) | 0.63 | (0.12, 0.85) | -1.5 [2.1] | (-5.7, 2.6) | -2.2 [1.9] | (-5.8, 1.4) | -1.3 [2.0] | (-5.2, 2.7) |
| 85% $VO_2$max | 0.63 | (0.03, 0.87) | 0.52 | (-0.10, 0.83) | 0.86 | (0.56, 0.95) | -2.1 [2.1] | (-6.3, 2.1) | -2.8 [2.0] | (-6.7, 1.1) | -0.9 [1.3] | (-3.4, 1.7) |
| HR | | | | | | | | | | | | |
| 60% $VO_2$max | 0.73 | (0.41, 0.89) | 0.68 | (0.30, 0.87) | 0.79 | (0.52, 0.91) | -0.3 [2.0] | (-4.2, 3.6) | -1.0 [1.8] | (-4.5, 2.5) | -0.5 [1.5] | (-3.4, 2.4) |
| 75% $VO_2$max | 0.70 | (0.37, 0.88) | 0.66 | (0.25, 0.86) | 0.81 | (0.57, 0.93) | -0.4 [2.3] | (-4.9, 4.1) | -1.2 [2.0] | (-5.2, 2.8) | 0.2 [1.6] | (-3.0, 3.3) |
| 85% $VO_2$max | 0.73 | (0.41, 0.89) | 0.64 | (0.12, 0.87) | 0.88 | (0.72, 0.95) | -0.9 [2.3] | (-5.5, 3.7) | -1.8 [2.1] | (-5.9, 2.4) | 0.3 [1.4] | (-2.4, 3.1) |
| ACT | | | | | | | | | | | | |
| 60% $VO_2$max | 0.02 | (-0.02, 0.12) | 0.04 | (-0.02, 0.18) | --- | --- | -8.5 [1.9] | (-12.3, -4.7) | -8.6 [1.7] | (-11.9, -5.3) | --- | --- |
| 75% $VO_2$max | 0.02 | (-0.01, 0.11) | 0.03 | (-0.01, 0.16) | --- | --- | -10.5 [2.1] | (-14.5, -6.5) | -10.5 [1.8] | (-14.0, -7.0) | --- | --- |
| 85% $VO_2$max | 0.03 | (-0.01, 0.14) | 0.03 | (-0.01, 0.17) | --- | --- | -11.8 [2.2] | (-16.0, -7.5) | -11.8 [2.0] | (-15.6, -7.9) | --- | --- |

Abbreviations: ICC, intraclass correlation coefficient; LoA, limits of agreement; AH, Actiheart; $AH_{2005}$, original group calibrated model termed "Group ACT/Group HR (old)" in the Actiheart software; $AH_{2007}$; latest group calibrated model termed "Group Cal JAP2007" in the Actiheart software; $AH_{2005}IND$, individually calibrated model termed "Group ACT/Ind HR (old)" in the Actiheart software; HR/ACT, HR and ACT, algorithm using respectively combined heart rate and activity data, only heart rate data and only activity data.

For the cycling exercise, when the HR/ACT or ACT algorithm was used, all group and individually calibrated models underestimated AEE as compared to IC (P-values <0.01; Fig 2D and 2F). This underestimation was much more pronounced for the ACT algorithm (Fig 2F). Conversely, the HR algorithm improved AEE estimation. Indeed, the underestimation of AEE measured by $AH_{2005}$ (P-values <0.001) was reduced for all intensities (Fig 2E, and bias in Table 2). For $AH_{2007}$, AEE was similar to IC for most intensities (P-values range: 0.06–0.27), except 85% $VO_2$max where a slight underestimation was still present (P<0.05). Following individual calibration ($AH_{2005}IND$), AEE estimated by the AH was still underestimated for the HR/ACT algorithm (P-values <0.01; Fig 2D), but was equal to IC for all intensities when the HR algorithm was applied (P-values range: 0.22–0.99; Fig 2E).

ICCs and bias are presented in Table 2. During both running and cycling, the mean bias increased with the intensity of exercise for the group calibrated models ($AH_{2005}$ and $AH_{2007}$), but not for the individually calibrated model ($AH_{2005}IND$; Table 2). When comparing the two

group calibrated models, except for the ACT algorithm during cycling, ICCs estimates and mean bias indicated a higher agreement with IC for $AH_{2007}$ compared to $AH_{2005}$ for all intensities. In contrast, the 95% LoA of bias were generally slightly larger for $AH_{2007}$ (Table 2).

For all intensities of the running exercise, whatever the model, ICCs, bias and 95% LoA were similar for the HR/ACT and HR algorithms. When the ACT algorithm was used, ICCs and bias of the two group calibrated models indicated a better agreement with IC for 60% $VO_2$max, a similar agreement for 75% and a lower agreement for 85% $VO_2$max compared to the HR/ACT and HR algorithms (Table 2). For the cycling exercise, ICCs, bias and 95% LoA indicated a higher agreement for the HR as compared to the HR/ACT algorithm, and a very low agreement for the ACT algorithm at all intensities (Table 2).

During both exercises, individual calibration greatly improved ICCs, bias and 95% LoA. It was however less effective for the cycling compared to running, as illustrated by the lower ICCs for $AH_{2005}$IND ($AH_{2005}$IND; Table 2), particularly when the HR/ACT algorithm was applied.

## Discussion

Since active subjects and high-level athletes are frequently involved in higher intensity exercises, the purpose of this study was to test the accuracy of the group calibrated models of the AH to quantify AEE during higher intensity running and cycling, and the additional benefit of an individual calibration of the HR-AEE relationship. Our results show that the AH is sensitive to each successive increase of exercise intensity when the HR or the combined HR/ACT algorithm is used. Group calibrated models however underestimate AEE for intensities ranging from 60 to 85% of $VO_2$max, even though the more recent model ($AH_{2007}$) slightly improves AEE estimation as compared to the original model ($AH_{2005}$). The best agreement with the reference method was obtained following individual calibration ($AH_{2005}$IND). For both group and individually calibrated models, accuracy of the HR algorithm was similar to the combined HR/ACT algorithm during running and was superior during cycling.

### Actiheart algorithms and accelerometer

Our results related to the ACT algorithm support the observations of several previous studies showing that accelerometers underestimate AEE of most physical activities, except walking and slow running [2,3,5,6,34,35]. The underestimation depends on exercise intensity and a plateau in AEE estimation has been reported for higher intensities [3,5,6]. Concomitantly, our results showed that AH HR and HR/ACT algorithms did not suffer from the ceiling effect reported for accelerometers, as indicated by an increase in the estimated AEE proportional to intensity increments up to 85% $VO_2$max.

Although it was suggested that combining ACT and HR data could improve AEE estimation during daily life activities and light to moderate intensity walking/running [17,18,20,23], our results do not support the superiority of the HR/ACT algorithm, as compared to the HR algorithm, during higher intensity exercises. The lack of improvement could be partly explained by the positioning of the AH accelerometer on the chest that does not allow to perceive lower limbs movements during cycling and attenuates acceleration signal during running [23,28] or to the uniaxial design [26–28,36]. In February 2019, the manufacturer released a fifth version of the AH device equipped with a tri-axial accelerometer. This fifth version however still operates with the algorithms developed for the uniaxial device. Therefore, further investigation is required to test the validity of the new device and to ascertain the real benefit of a chest mounted tri-axial accelerometer since it greatly shortens the maximal recording time.

Despite these limitations, our observation of a similar or greater accuracy of the HR algorithm for higher intensity running and cycling respectively, and the better accuracy reported for the combined algorithm for light to moderate intensity exercises suggest that a monitor enabling to choose the adequate algorithm for AEE analysis, according to activity type and intensity, represents an attractive option to improve AEE estimation during exercises of various intensities.

## Accuracy of group calibrated models at higher intensities

Most of the studies investigating the accuracy of the AH to estimate AEE mainly focused on low to moderate intensity exercises and rarely included very active subjects [22–24,28,34,35,37]. Compared to IC, they generally reported mean AEE under- or overestimations ranging from 3 to 20% for the group calibrated models using HR and HR/ACT algorithms during walking and jogging. Amongst those studies, the rare that investigated cycling exercises [22,23,34] generally reported a slightly higher error (8 to 28%).

Only two studies analyzed the accuracy of the AH during higher intensity exercises using a group calibrated model (version not specified) and the combined HR/ACT algorithm [26,27]. In addition, they limited their investigation to running. Koehler et al. [27] quantified AEE using IC and AH in men endurance athletes during an incremental running exercise which started at 10 km. h$^{-1}$ and ended at 17 km.h$^{-1}$ or at individual exhaustion. Running speed was increased by 1.4 km.h$^{-1}$ every 5 min with 30 s rest between each stage. Nichols and al. [26] tested the AH in female adolescent cross-country runners during three 8 min stages of treadmill running at individualized speeds corresponding to recovery, moderate and 5-km race speed. Both studies reported a significant underestimation of AEE ranging from 9 up to 36% for the higher running speeds. The underestimation reported in the present study was similar for the HR/ACT and HR algorithms and was lower (10 to 20%) than reported by the two previous studies. This discrepancy could be partly due to a difference in running economy [26] and the lower speed attained by our subjects during the highest stage. However, another factor that influences the estimation of AEE by the AH group calibrated models is the value set for the sleeping HR in the software. While Koehler et al. [27] clearly specified they were unable to determine sleeping HR for logistic reasons, Nichols et al. [26] did not mention this methodological aspect. Since the sleeping HR is lower in trained endurance athletes, this may also have contributed to the greater underestimation reported by the authors.

To our knowledge, our study is the first to investigate the accuracy of the AH to estimate AEE during high intensity cycling. Our results showed an underestimation for the group calibrated models ranging from 12 to 26% for the HR/ACT algorithm and from 3 to 12% for the HR algorithm. As compared to the running exercises at high intensity, this underestimation range is thus slightly greater for the HR/ACT algorithm and lower for the HR algorithm.

## Benefit of the individual calibration

AEE and the HR-AEE relationship depend on exercise type and different individual factors [9–12]. Besides subject's sex, these factors are not included as predictive variables of AEE in the AH group calibrated algorithms [20,32]. The only way to take them into account is to create new algorithms, which was not the aim of the present study, or to use individual calibration. In the present study, we therefore tested the benefit of the individual calibration of the HR-AEE relationship determined during a maximal cycling or running test.

Koehler et al. [27] reported that individual calibration of the HR-AEE relationship, using the built-in step test, reduces the mean underestimation in AEE by the AH, but the

95% LoA and validity correlation were not improved compared to the group calibration. In the present study, the individual calibration, based on the relationship between HR and AEE during the maximal running or cycling test, enables to considerably improve the agreement with IC for running and cycling exercises from 60 to 85% VO$_2$max (Fig 2 and Table 2). During running, based on the 95% confidence intervals of ICCs (Table 2), the agreement changed indeed from "poor to excellent" for group calibration to "excellent" for individual calibration [33]. During the cycling exercise, agreement was only slightly improved by individual calibration for the HR/ACT algorithm, whereas it was substantially better for the HR algorithm, passing from a "poor to moderate" to a "moderate to excellent" agreement (Table 2). The greater improvement in AEE when using a specific calibration as compared to the built-in step test calibration confirms the lack of accuracy of the step test in deriving the individual HR-AEE relationship for higher intensity exercises [27,36] and states the need for an individual calibration established during an exercise as close as possible to the sport and intensities performed by the subject. In practice, this calibration is suitable for athletes involved predominantly in one sport, and performing periodically maximal tests, but would be more complicated to implement for athletes taking part in different kind of sport activities.

Some considerations must be taken into account when interpreting our results and using the device for field measurements. Firstly, to ensure the feasibility of the individual calibration procedure and the precise determination of the load/speed corresponding to each intensity, our investigation was performed in a laboratory setting and was limited to running and cycling. Although these two physical activities are frequently practiced by most athletes and active subjects, we must acknowledge that our conclusions are not directly transposable to other sport activities. Secondly, our sample was composed of fit young subjects who are representative of the athletic population in whom an accurate assessment of AEE is needed for guidance on training volume and food intake. However, this observation warrants further studies to verify if the present conclusions are generalizable to individuals of different age and fitness levels. In addition, since body composition and physiological responses to exercise may differ between men and women [38] and influence AEE, a comparison of the accuracy of the AH group calibrated equations in subjects of both sex would be relevant. Finally, one practical problem encountered with the AH, in our and previous studies [21,36], is the loss of HR records in some subjects. During intense exercise, fast body movements and sweating may alter contact of ECG electrodes and induce ECG signal artifacts affecting the quality of HR data or leading to the loss of signal detection. This issue may limit the use of the AH in some athletes engaged in long sessions of high intensity exercise [36].

In conclusion, whilst taking into account the above-mentioned limitations, the present results indicate that the AH group calibrated models enable to estimate AEE during running and cycling exercises of higher intensity when the adapted algorithm is used and subject's sleeping HR is correctly set in the software. Combining HR and ACT does not appear to improve AEE estimation compared to using HR alone for higher intensities cycling and running. An individual calibration of the HR-AEE relationship, specific to the sport activity and intensity range, greatly improves AEE estimation. When a high accuracy in AEE quantification is needed, an individual calibration is therefore recommended.

## Supporting information

**S1 File. Study dataset.**
(XLSX)

## Acknowledgments

The authors would like to thank Jean-Charles Suaud and Romain Frayssinet for their help in data collection. This paper is published with the support of the University Foundation of Belgium.

## Author Contributions

**Conceptualization:** Malgorzata Klass, Vitalie Faoro, Alain Carpentier.

**Data curation:** Malgorzata Klass.

**Formal analysis:** Malgorzata Klass.

**Investigation:** Malgorzata Klass.

**Methodology:** Malgorzata Klass, Vitalie Faoro, Alain Carpentier.

**Project administration:** Malgorzata Klass, Vitalie Faoro, Alain Carpentier.

**Resources:** Malgorzata Klass, Vitalie Faoro, Alain Carpentier.

**Supervision:** Malgorzata Klass, Vitalie Faoro, Alain Carpentier.

**Validation:** Malgorzata Klass, Vitalie Faoro, Alain Carpentier.

**Visualization:** Malgorzata Klass.

**Writing – original draft:** Malgorzata Klass.

**Writing – review & editing:** Malgorzata Klass, Vitalie Faoro, Alain Carpentier.

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
