## [Decision Letter · Decision Letter 0]

3 Sep 2019

PONE-D-19-16566

Assessment of energy expenditure during high intensity cycling and running using a heart rate and activity monitor in young active adults

PLOS ONE

Dear Dr Klass,

Thank you for submitting your manuscript to PLOS ONE. After careful consideration, we feel that it has merit but does not fully meet PLOS ONE’s publication criteria as it currently stands. Therefore, we invite you to submit a revised version of the manuscript that addresses the points raised during the review process.

The article has merit and it is interesting. However, major changes should be made to improve the overall quality of the article before acceptance.

We would appreciate receiving your revised manuscript by Oct 18 2019 11:59PM. To enhance the reproducibility of your results, we recommend that if applicable you deposit your laboratory protocols in protocols.io, where a protocol can be assigned its own identifier (DOI) such that it can be cited independently in the future. For instructions see: http://journals.plos.org/plosone/s/submission-guidelines#loc-laboratory-protocols

We look forward to receiving your revised manuscript.

Kind regards,

Filipe Manuel Clemente, PhD

Academic Editor

PLOS ONE

Journal Requirements:

Reviewers' comments:

Reviewer's Responses to Questions

**Comments to the Author**

1. Is the manuscript technically sound, and do the data support the conclusions?

Reviewer #1: Partly

Reviewer #2: Yes

2. Has the statistical analysis been performed appropriately and rigorously? 

Reviewer #1: Yes

Reviewer #2: Yes

3. Have the authors made all data underlying the findings in their manuscript fully available?

Reviewer #1: Yes

Reviewer #2: Yes

4. Is the manuscript presented in an intelligible fashion and written in standard English?

Reviewer #1: Yes

Reviewer #2: Yes

5. Review Comments to the Author

Reviewer #1: Brief Summary:

Authors aimed to evaluate the accuracy of the Actiheart to assess activity related energy expenditure on HR and activity counts in young and very active subjects during running and cycling. To achieve this aim, they estimated total energy expenditure by using Weir equation and then compare it with data obtained from Activeheart based on different. However some methodological issues should be addressed. I would advise “major revision” for this manuscript based on general comments, as describe below:

General comments:

Abstract: the method for indirect calorimetry is not mentioned. In my opinion, it’s very important as this method is used as reference to verify the accuracy of the Actiheart data.

HR sleeping: I cannot understand the physiological relevance for the present study. Probably, its relevance is missing.

Could explain please why 2 individuals were excluded ? How data is lost? And the reliability of the data obtained from Actiheart ?

The main question of the present study is that you missed the determinants of energy expenditure that includes, body size, body composition, sex (as you used both, 4 female and 16male), age, physical fitness, hormonal status. How to ensure that there was no influence of these factors? Is any way to consider these factors in your data analysis ? Please, include a sentence in the introduction and discussion with information explaining how these determinants can influence your data ?

Reviewer #2: Abstract

Please add gender and age of the sample

Introduction

Would be better to add hypothesis

Methods

Participants

- Please add exclusion criteria

- Please explain how the activity level of the sample was determined

Protocol

- Please mention the time of the day of the experimental sessions

Sleeping HR

- Please add reference citation(s) to the sentence starting with “To that aim subjects had to …”

6. PLOS authors have the option to publish the peer review history of their article (what does this mean?). If published, this will include your full peer review and any attached files.

Reviewer #1: Yes: Sílvia Rocha-Rodrigues

Reviewer #2: No

---

## [Author Response · Author response to Decision Letter 0]

17 Oct 2019

Response to Reviewer #1: 

Brief Summary:

Authors aimed to evaluate the accuracy of the Actiheart to assess activity related energy expenditure on HR and activity counts in young and very active subjects during running and cycling. To achieve this aim, they estimated total energy expenditure by using Weir equation and then compare it with data obtained from Activeheart based on different. However some methodological issues should be addressed. I would advise “major revision” for this manuscript based on general comments, as describe below:

We thank the reviewer for the constructive comments that helped us to improve the clarity of the method section and the quality of the introduction and discussion. We did our best to better explain/discuss the points you raised and to adapt the manuscript as suggested. We hope to have met your expectations. Please find also below our responses to your comments.

General comments:

Abstract: the method for indirect calorimetry is not mentioned. In my opinion, it’s very important as this method is used as reference to verify the accuracy of the Actiheart data.

The method used is now clarified in the text, as requested (p2, line 28-29).

HR sleeping: I cannot understand the physiological relevance for the present study. Probably, its relevance is missing.

The group calibration HR algorithm incorporated in the Actiheart uses the HR above sleeping HR (HRaS) instead of “raw HR” to calculate AEE. Since sleeping heart rate varies from subject to subject and is lower in fit subjects (Andrews, 1971; Hiilloskorpi et al, 2003), using HRaS (based on individual sleeping HR) is thought to reduce the error due to between-individual variance in the HR–AEE relationship (Hiilloskorpi et al, 2003) used in the group calibration equations. A clarification has been added in the method section to explain the relevance (p8, line 136-139).

Could explain please why 2 individuals were excluded ? How data is lost? And the reliability of the data obtained from Actiheart ?

Based on previous literature, Actiheart HR data collection during walking and running is considered as reliable and similar to ECG recordings in laboratory environment (Brage et al 2005, Barreira et al 2009). However, during intense exercise, body movements and sweating may alter the contact of ECG electrodes and induce ECG signal artifacts affecting the quality of HR data or leading to the loss of signal detection. Poor ECG signal has generated unreadable HR data in 2 out of our 18 subjects. This clarification has been added in the methods (p6, line 97) and this limitation of the AH is now better explained in the discussion (p21, line 414-417).

The main question of the present study is that you missed the determinants of energy expenditure that includes, body size, body composition, sex (as you used both, 4 female and 16 male), age, physical fitness, hormonal status. How to ensure that there was no influence of these factors? Is any way to consider these factors in your data analysis ? Please, include a sentence in the introduction and discussion with information explaining how these determinants can influence your data ? 

We agree with the reviewer comment. All these determinants may influence energy expenditure during exercise since they impact balance of substrate utilization, work efficiency and the relation between HR increase and energy expenditure during exercise. There is however no clear consensus regarding the effect of the menstrual cycle on substrate utilization during exercise or on endurance performance (Hilloskorpi et al,1999, 20:438-4; Isacco et al, 2012 [Sports Med. 2012, 42:327-42]; Oosthuyse and Bosch, 2010 [Sports Med. 2010, 40:207-27]). 

Additional information has been added in the introduction (p10, line 63-67) and discussion (p19, line 377-382 and p21, 410-413) as requested.

The aim of the present study was not to evaluate the influence of those different parameters on energy expenditure during exercise but to validate the Actiheart with the user’s data and the models available in the Actiheart software and commonly used by practitioners and researchers. Data that need to be entered when a new user is created in the AH software are: age, sex, height and sleeping heart rate. Those parameters are used to estimate resting energy expenditure (using the Schofield equation; Schofield 1985). In addition to heart rate (value above sleeping heart rate) and activity counts, sex is also included in the HR and ACT Group calibration equations used by the Actiheart to estimate activity related energy expenditure (AEE). This precision has been added in the methods section (p10, line 192-194). Unfortunately, body composition, fitness level or other factors affecting energy expenditure are not taken into account in the group calibration equations. This partly explains why individual calibration of the HR-AEE relationship applied in the present study greatly improves energy expenditure estimation. 

Response to Reviewer #2: 

We thank the reviewer for the constructive comments that helped us to improve the clarity of the introduction and method section. We did our best to adapt the manuscript as suggested. We hope to have met your expectations. Please find also below our responses to your comments.

Abstract

Please add gender and age of the sample

Gender and age have been added as requested (p2, line 28).

Introduction

Would be better to add hypothesis

Hypothesis has been added in the introduction as requested (p5-6, line 91-93).

Methods

Participants

- Please add exclusion criteria

Inclusion and exclusion criteria have been added as requested (p6, line 99-101).

- Please explain how the activity level of the sample was determined

No specific method (questionnaire, agenda or other) was used to quantify the activity level of the sample. The activity level was estimated by the answers to the question: “Which sport(s) do you practice and how many times a week?” This part of the method section has been adapted to make it clearer we did not use a specific method to quantify the activity level of the sample (p6, line 106-107). 

Protocol

- Please mention the time of the day of the experimental sessions

This is now clarified in the methods (p6, line 111). For each subject, the experiments were conducted at about the same time in the morning or the afternoon. We also added that subjects had to refrain from food and caffeine intake within the 3 h before the experiment (p6, line 120). 

Sleeping HR

- Please add reference citation(s) to the sentence starting with “To that aim subjects had to …”

The relevance of the sleeping HR and a reference has been added as requested (p8, line 136-139 and 143).

---

## [Decision Letter · Decision Letter 1]

25 Oct 2019

Assessment of energy expenditure during high intensity cycling and running using a heart rate and activity monitor in young active adults

PONE-D-19-16566R1

Dear Dr. Klass,

We are pleased to inform you that your manuscript has been judged scientifically suitable for publication and will be formally accepted for publication once it complies with all outstanding technical requirements.

With kind regards,

Filipe Manuel Clemente, PhD

Academic Editor

PLOS ONE

Additional Editor Comments (optional):

Reviewers' comments:

Reviewer's Responses to Questions

**Comments to the Author**

1. If the authors have adequately addressed your comments raised in a previous round of review and you feel that this manuscript is now acceptable for publication, you may indicate that here to bypass the “Comments to the Author” section, enter your conflict of interest statement in the “Confidential to Editor” section, and submit your "Accept" recommendation.

Reviewer #1: All comments have been addressed

Reviewer #2: All comments have been addressed

2. Is the manuscript technically sound, and do the data support the conclusions?

Reviewer #1: Yes

Reviewer #2: Yes

3. Has the statistical analysis been performed appropriately and rigorously? 

Reviewer #1: Yes

Reviewer #2: Yes

4. Have the authors made all data underlying the findings in their manuscript fully available?

Reviewer #1: Yes

Reviewer #2: Yes

5. Is the manuscript presented in an intelligible fashion and written in standard English?

Reviewer #1: Yes

Reviewer #2: Yes

6. Review Comments to the Author

Reviewer #1: The authors exceeded all the questions and suggestions of reviewers. Based on that, I recommend accept for this publication.

Reviewer #2: (No Response)

7. PLOS authors have the option to publish the peer review history of their article (what does this mean?). If published, this will include your full peer review and any attached files.

Reviewer #1: Yes: Sílvia Rocha-Rodrigues

Reviewer #2: No

---

## [Editor Report · Acceptance letter]

29 Oct 2019

PONE-D-19-16566R1 

Assessment of energy expenditure during high intensity cycling and running using a heart rate and activity monitor in young active adults 

Dear Dr. Klass:

I am pleased to inform you that your manuscript has been deemed suitable for publication in PLOS ONE. Congratulations! Your manuscript is now with our production department. 

With kind regards,

on behalf of

Dr. Filipe Manuel Clemente 

Academic Editor

PLOS ONE